# New insights into the criteria of functional heterozygosity of the *Apis mellifera* complementary sex determining gene–Discovery of a functional allele pair differing by a single amino acid

**Robert Mroczek[1], Agnieszka Laszkiewicz[1], Pawel Blazej[2], Kinga Adamczyk-Weglarzy[1], Joanna Niedbalska-Tarnowska[1], Malgorzata Cebrat**[1] *

1 Laboratory of Molecular and Cellular Immunology, Ludwik Hirszfeld Institute of Immunology and Experimental Therapy Polish Academy of Sciences, Wroclaw, Poland, 2 Department of Bioinformatics and Genomics, Faculty of Biotechnology, Wroclaw University, Wroclaw, Poland

* malgorzata.cebrat@hirszfeld.pl

**Data Availability Statement:** All csd allele sequences identified in this work are available from

## Abstract

The complementary sex determiner (*csd*) gene is responsible for controlling the sex-determination molecular switch in western honey bees (*Apis mellifera*): bees that are heterozygous for *csd* develop into females, whereas bees that are hemizygous or homozygous develop into males. The homozygous diploid males are destroyed at an early stage of their development. It has been proposed that the minimal number of amino acid differences between two *csd* alleles needed to fully determine femaleness is five and it has also been shown that smaller differences may result in forming an evolutionary intermediate that is not fully capable of female determination, but has increased fitness compared to the homozygous genotype. In this study, we have implemented a terminal restriction length polymorphism-based method of identifying and distinguishing paternal alleles in a given bee colony and assigning them to a particular maternal allele in order to gather information on large number of functional *csd* pairs and also to identify, to some extent, genotypes that are underrepresented or absent in bee colonies. The main finding of this study is the identification of a fully functional genotype consisting of *csd* alleles that differed from each other by a one amino acid position. The individuals carrying this genotype expressed only female-specific transcripts of feminizer and double-sex genes. By comparing the sequences differences between the *csd* pair identified in our study with those described earlier, we conclude that functional heterozygosity of the *csd* gene is dependent not only on the number of the amino acid differences but also on the sequence context and position of the change. The discovery of a functional allele pair differing by a single amino acid also implies that the generation of a new *csd* specificity may also occur during a single mutation step with no need for evolutionary intermediates accumulating further mutations.

the NCBI database (accession numbers MZ030644-MZ0306717).

**Funding:** The research was supported by National Science Centre, Poland (grant No UMO-2018/31/B/NZ9/00969) awarded to MC. The funders had no role in study design, data collection and analysis, decision to publish, or preparation of the manuscript.

**Competing interests:** The authors have declared that no competing interests exist.

## Introduction

Western honey bees (*Apis mellifera*) are haplodiploid: the males develop from unfertilised oocytes, whereas the females develop from fertilised oocytes. The complementary sex determiner (*csd*) gene is responsible for controlling the sex-determination molecular switch [1]: bees that are heterozygous for *csd* develop into females, whereas the female developmental pathway is not activated in bees that are hemizygous or homozygous, which leads to default male development. The homozygous diploid males are eaten by worker bees shortly after they hatch from the egg [2]. The risk of homozygous bee formation is reduced by polyandry (in natural conditions, the queen mates with several drones), behavioural traits limiting the possibility of sib-mating, and most importantly, a high number of *csd* alleles present in the population [3]. The lethality of homozygous bees is the reason why new alleles appearing in the population are highly favoured: the probability that a rare allele forms a homozygous pair is low, and therefore, the *csd* gene evolves under balancing (negative frequency-dependent) selection [4]. It has been shown that the total number of *csd* alleles is difficult (or even impossible) to estimate due to the uneven distribution of the alleles and the occurrence of a large number of infrequent alleles [5]. Currently, publicly available databases contain several hundred different *csd* sequences [6, 7].

Such a high diversity of *csd* alleles is possible due to the specific structure of the *csd* gene. The gene consists of 9 exons, out of which exons 6–8 encode the potential-specifying domain (*csd*-PSD), which has been identified as a target for balancing selection [1, 8]. The *csd*-PSD itself contains a hypervariable region (HVR) flanked by arginine/serine- and proline-rich regions, which are likely responsible for protein-protein interactions (Fig 1). The HVR, which encodes the tyrosine- and asparagine-rich portion of the *csd*-PSD, is A/T-rich and contains several trinucleotide repeats (TAT and TAA), often arranged as sets of motifs. It has been

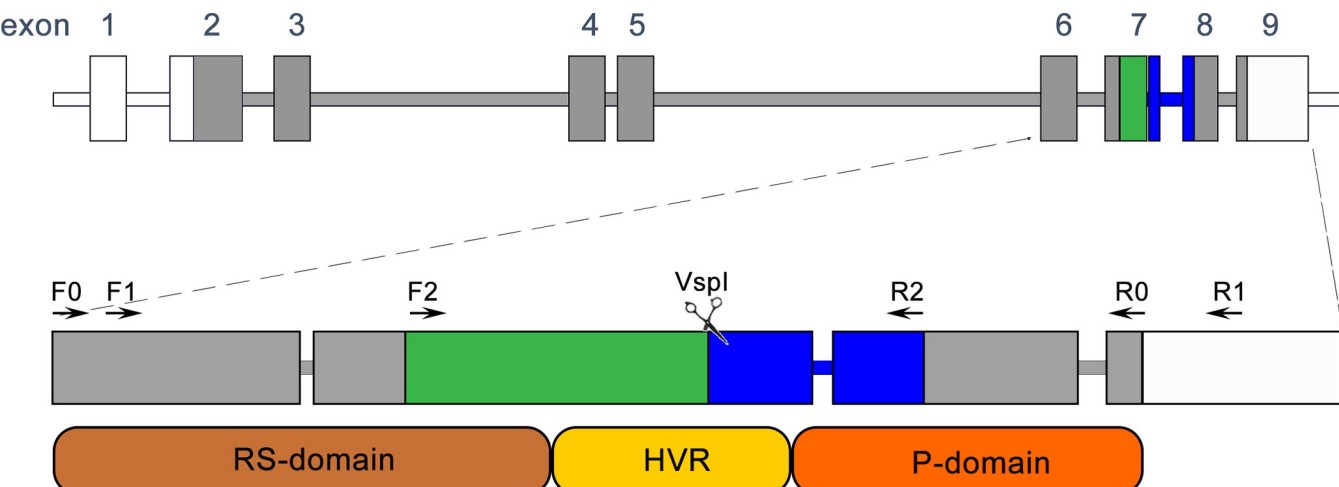

**Fig 1. Organisation of the *Apis mellifera csd* gene locus.** Rectangles represent exons and white exons or their fragments represent non-coding parts of the gene. Arrows represent the position of primers used in this study. The green and blue fragments of exons 7 and 8 represent the localization of the amplicon and its restriction fragments analysed by the T-RFLP method used in this work. The amino acid sequence of an example of a potential-specifying domain of the csd protein (csd-PSD) is given together with the localization of its arginine/serine-rich domain (RS-domain, brown), hypervariable region (HVR, yellow) and proline-rich domain (P-domain, orange).

proposed that the microsatellite-like structure of the HVR is responsible for its high mutation rate, most likely due to polymerase slippage during DNA replication and/or unequal crossing-over. Differences in the length and composition of the HVR are the major contributing factor to the diversity of the *csd* alleles. This diversity is further increased due to the occurrence of nucleotide substitutions outside the HVR. In contrast to *csd*-PSD, the gene fragment encoding the N-terminal part of the csd protein accumulates far fewer mutations and has been proven to be a target of purifying selection [4].

The occurrence of different *csd* alleles in the embryo leads to female-specific splicing of the feminizer (*fem*) transcript, which results in the presence of a functional fem protein, which in turn leads to female-specific splicing of the doublesex (*dsx*) transcript encoding a version of the protein truncated at the C-terminus, which initiates the cascade of female development. The lack of different *csd* alleles or a complete absence of *csd* gene products results in male-specific *fem* splicing, which encodes a non-functional protein. Lack of the fem protein leads to the appearance of a variant of the doublesex transcript and protein that determine male development [1, 9, 10]. Although it has not yet been proven, it is likely that csd acts as a protein dimer that can be formed or is functional only as a heterodimer; however, the exact molecular mechanism of csd action remains to be established [11]. Another major issue is determining the minimal difference in the *csd* sequence(s) that is sufficient for establishing functional heterozygosity. After analysing differences in *csd* alleles occurring in females, Lechner et al. proposed the following criteria: $d_{HVR} \geq 6$, $d_{PSD} \geq 1$ and $3_{dPSD} + 2_{de8} \geq 9$, where: $d_{HVR}$ is the difference in the length of the HVR; $d_{PSD}$ is the number of amino acid mismatches in the PSD region; and $d_{e8}$ is the number of amino acid mismatches in the part of the protein that is encoded by exon 8 [12]. These criteria were challenged by a subsequent report proposing that at least 5 amino acid differences and length variations in the *csd*-PSD are able to regularly induce femaleness [13]. The same report described a case of the co-occurrence of a pair of *csd* alleles encoding proteins that differed by only 3 amino acids in the HVR region. This pair of alleles predominantly induced lethality, but in infrequent cases, also femaleness. The incomplete penetrance of femaleness has been suggested as a mechanism through which new *csd* specificities can gradually evolve.

However, because only one case of a genotype consisting of non-identical *csd* alleles unable to fully establish femaleness has been identified so far [13], establishing any conclusive criteria for functional heterozygosity is difficult or even impossible. In this study, we implemented an approach enabling the analysis of a large number of diploid *csd* genotypes to gather information on functional *csd* pairs and also to identify, to some extent, genotypes that are underrepresented or absent in bee colonies in order to verify/refine the criteria of functional heterozygosity of *csd* alleles. The main finding of this study is the identification of a fully functional genotype consisting of *csd* alleles that differed from each other by a one amino acid position.

## Results

### Rationale and workflow of the genotyping method

Defining the criteria for the functional heterozygosity of *csd* alleles requires, on the one hand, gathering information on existing functional *csd* genotypes, i.e. those present in viable females, and on the other hand, learning about allele combinations that are 'forbidden', i.e. do not form functional pairs. The latter should not be present in a given population, since they lead to the formation of non-viable diploid males or, as in the case described by Beye et. al., should be strongly underrepresented in the female population, as those able to determine femaleness but with incomplete penetrance [13]. Prior to sequencing, the *csd* alleles derived from the females would have to be cloned, which makes this approach impossible to carry out on a larger scale.

On the other hand, the high-throughput approaches which have been described recently albeit excellent for identifying *csd* alleles present in the population, do not give information on their functional combinations i.e. which alleles are able to form pairs determining femaleness [7]. Consequently, we developed an approach that took advantage of the fact that *csd* alleles most frequently differ from each other in the number of trinucleotide repeats in the HVR, as well as the TTCCTG/A repeats in the proline-rich encoding region of the gene. In virtually all *csd* allele sequences analysed to date, these two parts are separated by a conserved sequence containing the *VspI* restriction enzyme recognition site (ATTAAT) (Fig 1). These features of the *csd* sequence allowed us to develop a T-RFLP (terminal restriction length polymorphism)-based method of identifying and distinguishing paternal alleles in a given bee colony and assigning them to a particular maternal allele. The aim was to a) identify *csd* allele combinations present in females, b) identify (whenever possible due to the appropriate number of analysed individuals within the patriline and the fulfilment of the criteria for statistical tests) significant biases in the frequency of genotypes consisting of a given paternal allele and the two maternal alleles present in a given colony. We assumed that the distribution of the maternal alleles within the group of worker bee pupas carrying the same paternal allele would be 0.5/0.5 unless one of the genotypes is lethal or partially lethal; if identified, the absent/underrepresented genotypes should be the ones that are unable to fully determine femaleness. Here we would like to stress that by implementing this method we did not attempt to analyse and compare the frequencies of occurrence of paternal *csd* alleles at the level of the entire colony but only compare the frequency by which a given paternal allele pairs with the maternal alleles.

A potential drawback of the above-proposed method is that the number of restriction patterns is smaller than the number of different *csd* sequences, and therefore, assigning paternal alleles to individual groups based solely on their restriction patterns may lead to an incorrect merging of two or more distinct paternal alleles into a single group. In order to identify the extent of this problem, we created and analysed a virtual dataset beforehand, consisting of *csd* sequences present in publicly available databases. We extracted the records that contained the entire sequence in question and identified *in silico* the *VspI* restriction patterns. The final dataset contained sequences representing 151 different *csd* alleles and 79 restriction patterns (S1 File). From this set, we created subsets containing 80 randomly-picked distinct records—number of records in the subsets was chosen based on the reported *csd* diversity present in the local populations [5]. Assuming that a queen bee mates with 10–20 drones, we created 'virtual spermathecas' by sampling with replacement 10, 12, 14 16, 18 and 20 random records from each subset and counted the number of different alleles and distinct restriction patterns in the obtained samples (Table 1). The sampling was repeated 10 times for each subset and

**Table 1. The comparison of the number of different *csd* alleles and their *VspI* restriction patterns in virtual spermathecas of different sizes (containing 10–20 records/"alleles").**

| size of the spermatheca | mean number of different alleles | mean number of restriction patterns |
|:---:|:---:|:---:|
| 20 | 17.7 | 16 |
| 18 | 16.2 | 14.7 |
| 16 | 14.5 | 13.4 |
| 14 | 12.9 | 11.9 |
| 12 | 11.2 | 10.6 |
| 10 | 9.5 | 9 |

The numbers are mean values obtained after drawing with replacement (n = 10) a specified by the spermatheca size number of records from 80-record subsets (n = 10) of a dataset containing 151 different *csd*-PSD sequences available in the public repositories.

spermatheca size. Thus, we obtained an approximation of the expected number of different alleles in all analysed cases. In the case of the 20 represented sampled records, we obtained 17.7 different alleles and 16 different restriction patterns. Smaller differences between the respective mean numbers were observed when a small 'spermatheca' size was used.

We concluded that the possible error generated by the proposed screening method is small enough to warrant its further application, especially that a merged group of distinct paternal alleles could easily be distinguished as the one containing significantly more elements than the other groups. Alternatively, such large groups can also represent those containing the offspring of multiple drones possessing identical *csd* alleles. However, the two possibilities are indistinguishable in this method.

The method consisted of the following steps: 1. Identification of maternal alleles through genotyping of the drones present in a given colony: the genotyping involved PCR amplification of the portion of the *csd*-PSD using 6FAM- and HEX-labelled primers followed by identification of the lengths of both restriction fragments obtained after *VspI* digestion and direct sequencing of the amplicons. 2. Amplification of the *csd* alleles from several hundred worker bee at the pupa stage present in a given colony individually and subsequent T-RFLP analysis of the amplicons. Using the previously obtained information on the restriction pattern of the maternal alleles present in the colony, the restriction pattern of the paternal allele was identified. 3. Assigning the worker bees into groups based on the restriction pattern of the paternal allele, and within the particular groups, to one of the two maternal alleles. 4. Cloning and sequencing of the paternal *csd* alleles representing particular groups. 5. a) Aligning the sequences of *csd* alleles present in the identified genotypes and b) identifying potential statistically significant differences in the frequencies of the pairing of a given paternal allele with the respective maternal alleles and analysis of the sequence differences between the maternal and paternal alleles within the identified genotypes. The overview of the method is presented in Fig 2.

### Identification and characterisation of *csd* genotypes

We have tested worker bee pupas from five colonies led by naturally inseminated one-year-old queens. First, we identified the *VspI* restriction pattern and sequenced the maternal alleles

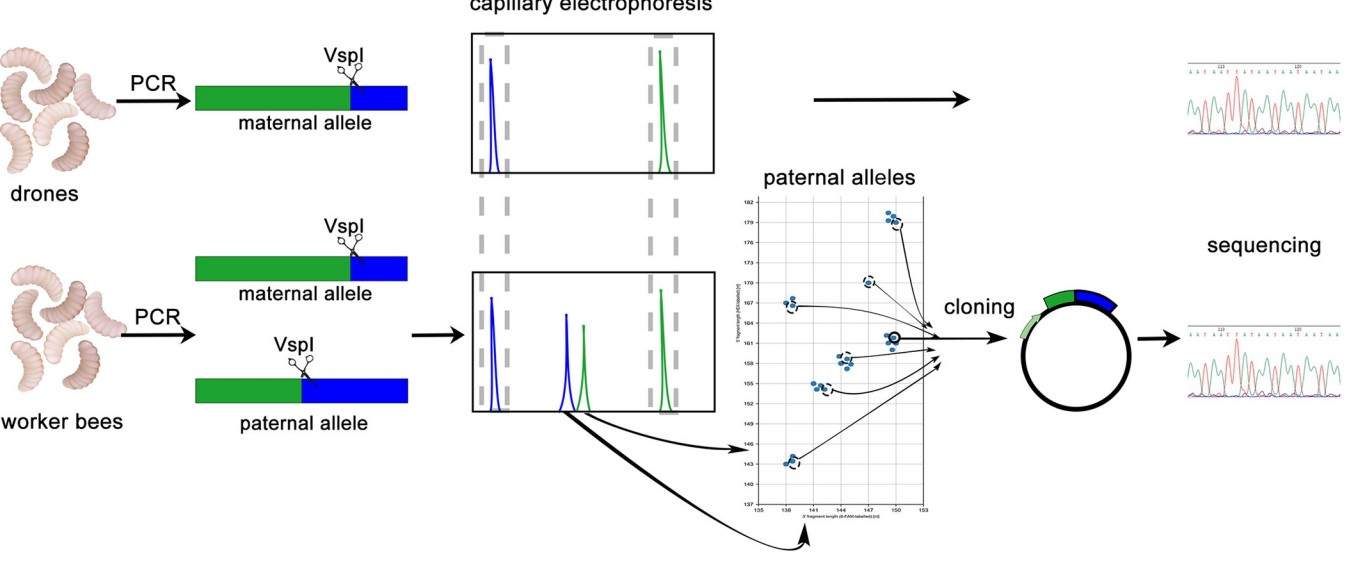

**Fig 2. Schematic representation of the methodology of identifying csd genotypes in worker bees.**

present in a given colony by drone genotyping, identifying 7 distinct alleles in total. Several hundred worker bees were then genotyped and assigned to groups based on the presence of a given paternal allele identified through the unique restriction pattern (Fig 3A). An example of the restriction pattern map of the paternal alleles of one of the colonies is shown in Fig 3B. In total, we established the maternal and paternal *csd* restrictions patterns of 1248 worker bees, identifying from 7 to 18 paternal alleles in a single colony. As a result of the cloning and sequencing of the paternal alleles of the representatives of each group, we identified 54 distinct paternal alleles and 118 distinct *csd* genotypes (Table 2 and S2 File, accession numbers MZ030644-MZ0306717).

Next, we have aligned the sequences of *csd* allele pairs forming the identified genotypes. As a result, we have created a two-dimensional heat map representing the sequence differences (Fig 4); since the nature of the HVR region does not allow for reliable sequence alignment, the differences between the *csd* alleles forming a given genotype were characterised by two parameters: the difference in the length of the HVRs (given by the number of amino acid residues) and the number of amino acid differences (mismatches and indels) outside of the HVRs (the fragments of the RS- and proline-rich domains) [13]. As shown in Fig 4, the differences in the length of HVRs ranged from 0 to 16, whereas the number of amino acid differences outside the HVRs ranged from 0 to 12 (with the median values equal to 6 and 5, respectively). Then we have analysed the frequencies with which the given paternal alleles formed a pair with the two maternal alleles present in a given colony (Fig 3C and S3 File). At this point it is important to note that a large number of patrilines (33) had to be excluded from the analysis due to an insufficient number of individuals (n<16) assigned to a given patriline–this was most probably due to the naturally occurring biases in the patriline composition reflecting the actual contribution of a particular drone to the number of spermatozoa stored in the spermatheca and insufficient mixing of sperm at the beginning of ovi-position [14]. While most of the remaining patrilines were characterised by nearly equal frequencies of the presence of maternal alleles, we were able to identify four cases that were characterised by different frequencies of the pairing of the paternal allele with the maternal alleles (chi-test, p-value adjusted for multiple testing < 0.1). We assumed that these uneven frequencies were caused by the partial lethality of these underrepresented *csd* genotypes, and compared the sequence differences between the *csd* alleles in such genotypes to the other genotypes identified in our study using the aforementioned map of sequence differences. We found that the sequence differences between the *csd* alleles forming the genotypes underrepresented in our analysis (Fig 4, red dots) were indistinguishable from the differences characterising other genotypes.

More importantly however, we have investigated the *csd* genotypes characterised by sequence differences smaller than the assumed minimal difference necessary to regularly induce femaleness (five amino acid differences and length variations in the *csd*-PSD [13]). We initially identified 11 such genotypes (Fig 4); however, nine of them, apart from small differences in the length of the HVR region, were characterised by several amino acid mismatches in this region. The remaining two genotypes, consisting of mB-p66 and mB-p29 *csd* alleles (where "m" and "p" stands for maternal and paternal alleles, respectively; B–the identifier of the maternal allele; the numbers–identifiers of the paternal alleles [see S2 File for details]), were characterised by a total of one and four amino acid differences in the analysed fragment of the csd-PSD, respectively (Fig 5). These two genotypes were subject to further analysis.

## Analysis of the fully functional *csd* pairs with a minimal sequence difference

The paternal p66 allele was present in 22 worker bees in one of the colonies (3–2019), out of which 13 specimens carried the above-mentioned nearly identical mB-p66 csd genotype and 9

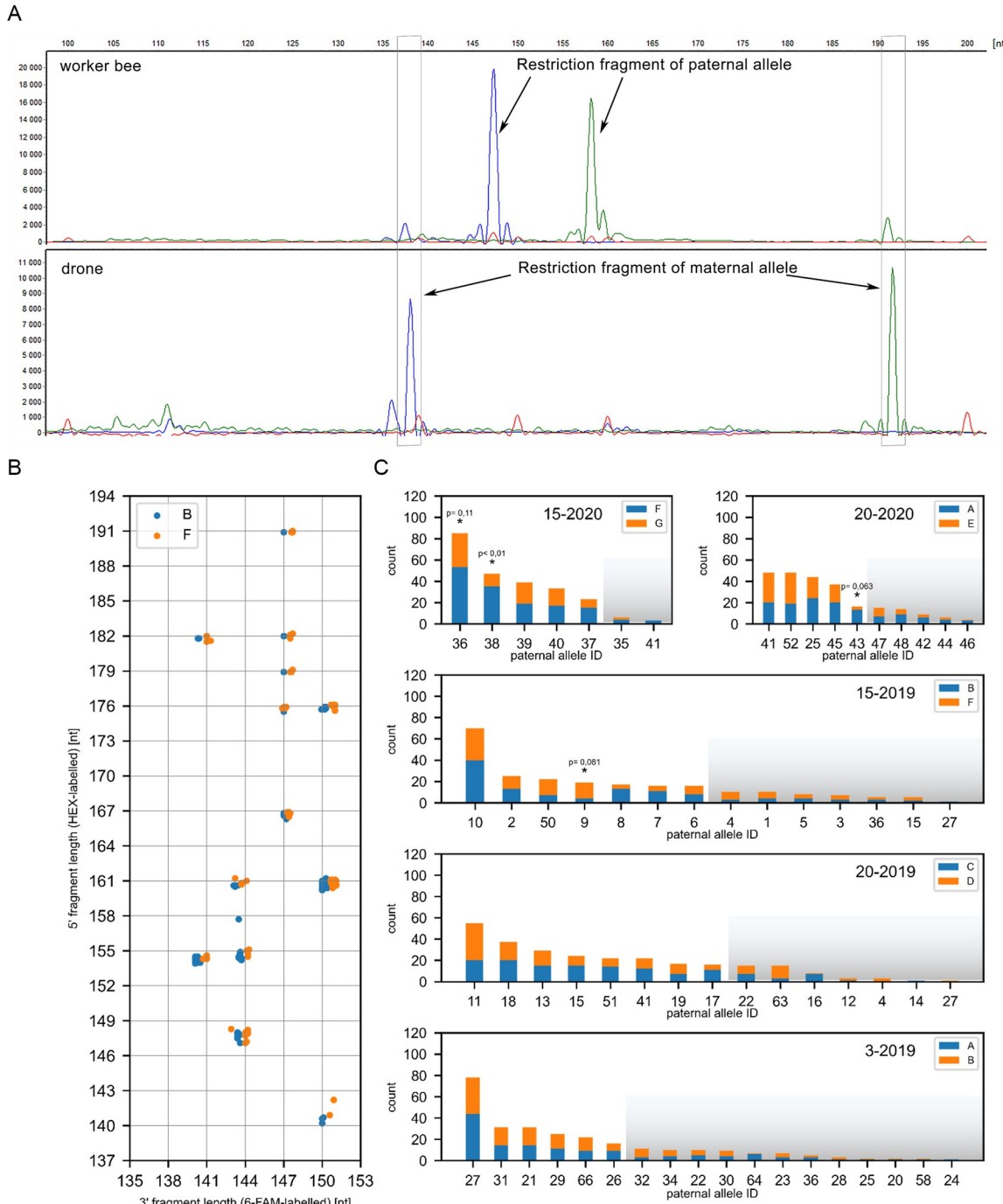

**Fig 3. Terminal restriction fragment length polymorphism method of identifying paternal *csd* alleles in bee colonies.** (A) An example of T-RFLP analysis of a drone (maternal allele) and a worker bee (identification of a paternal allele) *csd* alleles. (B) An example of the distribution of the paternal alleles based on the length of the *VspI* restriction fragments of analysed portion of the *csd* gene assessed by the T-RFLP method (colony 15–2019). Blue and orange dots represent paternal alleles paired with a given maternal alleles. (C) Frequency of the occurrence of the paternal alleles in each colony and the distribution of the maternal alleles (blue and orange bars) in each *csd* patriline. Paternal allele groups with statistically significant (chi-test with Bonferroni correction for multiple testing, p<0.1) differences in pairing with maternal alleles are marked with asterisk. Shaded parts of the graph indicate patrilines that were excluded from the analysis due to too small number of individuals in a given patriline.

**Table 2. Summary of the identified *csd* alleles and genotypes in the analysed colonies.**

| colony identifier | 3–2019 | 15–2019 | 20–2019 | 15–2020 | 20–2020 |
|---|---|---|---|---|---|
| number of analysed worker bees | 272 | 231 | 268 | 236 | 241 |
| frequency of maternal alleles[1] | A:134, B:138 | B:116, F:115 | C:133, D:135 | F:146, G:90 | A:125, E:116 |
| number of paternal alleles[*] | 18 | 14 | 15 | 7 | 10 |
| number of patrilines with different frequency of maternal alleles[#] | 0 | 1 | 0 | 2 | 1 |
| total number of distinct maternal alleles | 7 | | | | |
| total number of distinct paternal alleles | 54 | | | | |
| total number of distinct genotypes | 118 | | | | |

[1] number of worker bees within a colony carrying a given maternal *csd* allele

[*] number of the identified distinct paternal *csd* restriction patterns

[#] statistically significant (chi-test, p-value adjusted for multiple testing < 0.1) differences in frequencies of occurrence of maternal *csd* alleles in a given *csd* patriline.

carried the mA/p66 genotype. The mB and p66 alleles differed from each other by a trinucleotide indel (AAT) encoding asparagine residue localised in the hypervariable portion of the protein. The difference between p66 and the other maternal allele (mA) was more profound (a 3 a-a difference in the HVR length, 5 other a-a mismatches in HVR, and 3 a-a differences in the P-domain) (Fig 5). In order to exclude the possibility that the p66 allele was an artefact that had been accidentally derived from the mB allele as a result of a polymerase error during the amplification stage of genotyping, the *csd* alleles of several specimens assigned to the p66

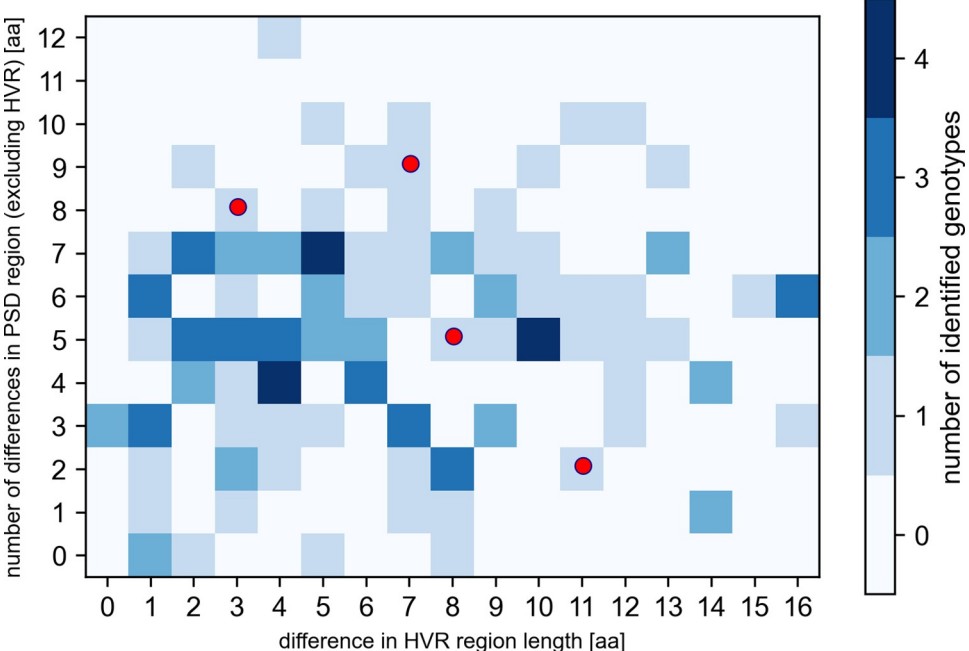

**Fig 4. The distribution of sequence differences between maternal and paternal *csd* alleles in the identified genotypes.** The diagram represents the differences between maternal and paternal csd-PSD sequences in genotypes detected in this study characterized by the difference in HVR (amino acid) length and the number of amino acid differences outside of the HVR (X- and Y-axis, respectively). Different shadings depict the number of genotypes characterized by a given set of parameters. Red dots represent the genotypes identified as underrepresented in the analysed colonies.

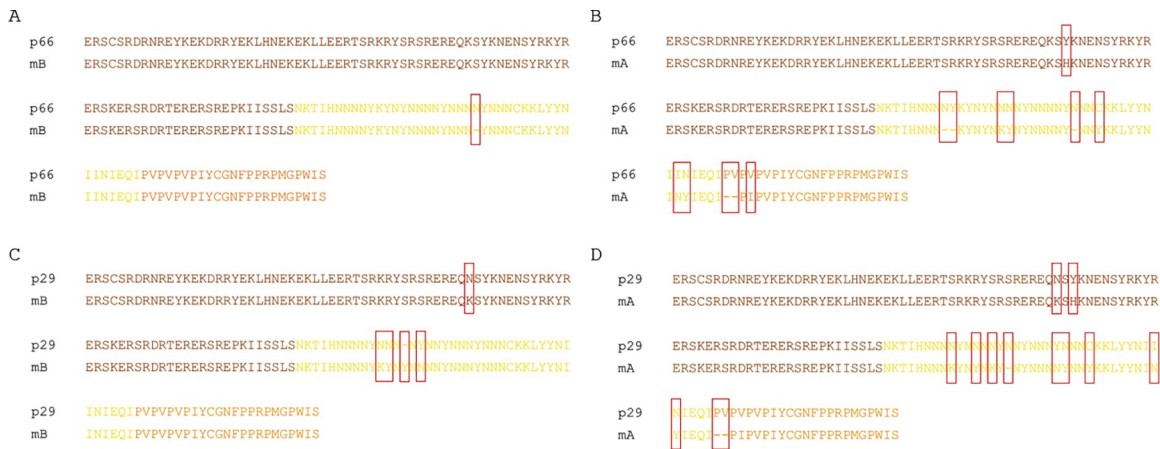

**Fig 5. Alignment of amino acid sequences of csd-PSD of p66/mB, p66/mA, p29/mB and p29/mA alleles.**

paternal allele group were cloned and sequenced and were found to have an identical p66 allele as the originally identified, paired either with the mB or mA maternal allele. The sequences of the maternal alleles were independently verified using drone genotyping. We then sequenced an extended portion of p66 and mB alleles to cover the protein-coding sequence present in exons 6–8. We found one C>T substitution localised in exon 6; however, this change did not change the sequence of the encoded protein. To exclude the possibility that we have accidently amplified *csd* pseudogenes instead of the functional genes we have confirmed the expression of the csd-PSD region of both mB and p66 alleles by RT-PCR using cDNA obtained from mB-p66 larvae. The T-RFLP analysis and sequencing of the cloned amplification products showed the lack of the intron between exons 7 and 8 whereas the rest of the sequence fragments were identical to the ones identified during the analysis of the genomic DNA. It is also important to mention that we have been unable to obtained such intronless amplicons during the amplification of the genomic DNA–this excludes the possibility of the presence of the processed pseudogenes in the genome.

Although the frequency of the occurrence of the mB-p66 genotype vs. the mA-p66 genotype (13/9) strongly indicated that the mB-p66 genotype did not cause lethality, we nonetheless tested whether bee larvae carrying the mB-p66 genotype had the molecular hallmark of female development. This was done in order to exclude the possibility that the mB-p66 genotype leads to the formation of diploid males, which are not recognised as such by the worker bees. Consequently, we tested the mB-p66 larvae for the presence of female and/or male splice variants of the *dsx* transcript using RT-PCR. In all tested cases, the mB-p66 larvae expressed only the female-specific *dsx* transcript (Fig 6, S1 Raw images).

We used the same method to analyse the second pair of alleles, mB-p29. This pair was characterised by one amino-acid indel in the HVR region accompanied by three other amino-acid mismatches in the HVR. Again, the sequence difference between p29 and the other maternal allele (mA) was much larger (1 a-a difference in the length of the HVR, 8 other a-a mismatches in the HVR and 3 a-a differences in the P-domain). The number of occurrences of the mB-p29 and mA-p29 genotypes in the analysed colony was 14 and 11, respectively, showing no indication that mB-p29 genotype may cause lethality. Similar to mA-p29, the mB-p29 individuals expressed a female splice variant of the *dsx* transcript (Fig 6, S1 Raw images). However sequencing of the additional portion of the RS-domain revealed additional amino acid substitution (K>N, Fig 5).

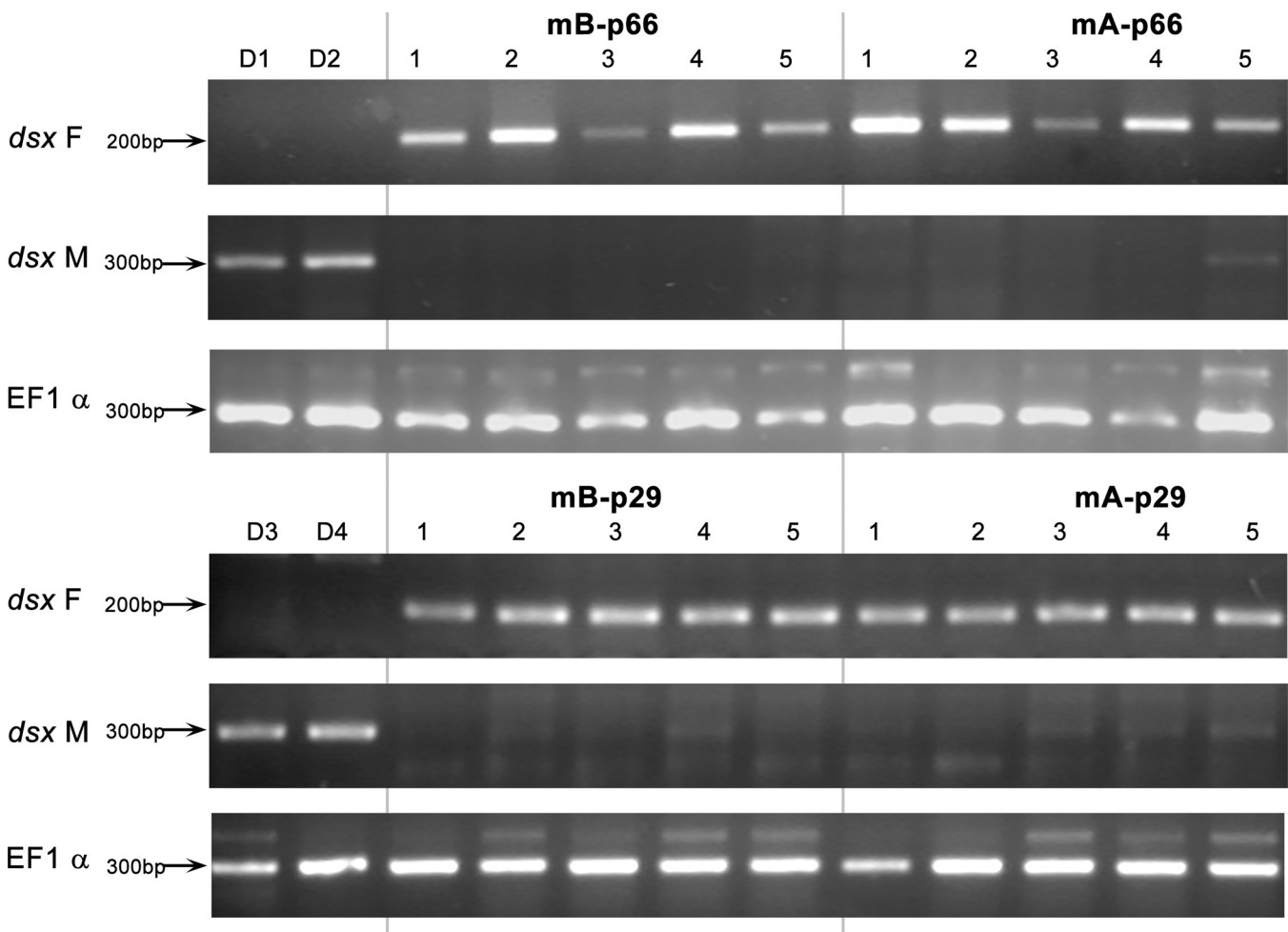

**Fig 6. RT-PCR analysis of female- (dsxF) and male-specific (dsxM) splice variants expression of *dsx* gene in individuals possessing mB-p66 and mA-p66 and mB-p29 and mA-p29 genotypes.** Numbers 1–5 depict randomly chosen representants of each genotype, D1-4: drone individuals used as controls. Amplification of EF1α gene was used as a positive control.

## Discussion

The extreme diversity of the honeybee *csd* gene (several hundred of its alleles have been identified to date) means that the number of theoretically possible genotypes heterozygous for *csd* gene is enormous. However, the number of functional heterozygous genotypes actually present in local populations can safely be assumed to be smaller, for two reasons: firstly, the *csd* alleles show an uneven spatial distribution across the honey bee population, making the number of available variants in local populations limited [5]; secondly, not all alleles can be expected to have the ability to form functional pairs with each other due to insufficient sequence differences.

Establishing the criteria of functional heterozygosity is difficult for many reasons, the most important of which is the lack of knowledge about the detailed mechanism of action of the csd protein and its interaction with itself (formation of homo- and/or heterodimers) and with other proteins, which makes it impossible to focus the analysis on precisely selected regions of the csd protein, e.g. by creating an *in vitro* model and site-directed mutagenesis. An obstacle which in turn prevents the development of a computer algorithm that would predict the ability

of a given pair of alleles to form a functional pair is an almost complete lack of data on the pairs of alleles that differ in sequence, but are not functional and incapable (or not fully capable) of determining female development. Therefore, it is not surprising that along with identifying new *csd* genotypes, the predicted minimum sequence difference continues to decrease [12, 13, 15] along with identifying new genotypes. In this study, we have implemented a method allowing for efficient screening of bee colonies in order to identify *csd* alleles and their combinations present in female individuals. Instead of randomly picking worker bees in order to clone and sequence their *csd* alleles we have segregated the analysed bees into patrilines using T-RFLP method, thus greatly reducing the number of individuals needed to establish the sequence of *csd* alleles present in the colony. While we found this method very efficient in identifying diploid *csd* genotypes it has not been that efficient in identifying potential biases in the frequencies with which a given paternal allele pairs with the maternal alleles—approx. 50% of the patrilines had to be excluded from the analysis due to the insufficient number of individuals needed to draw statistically meaningful results. Most likely this problem is a result of naturally occurring differences in the contributions of particular drones to the number of spermatozoa present in the spermatheca and it can be resolved by increasing the number of analysed worker bees; however, the actual extent of the bias and therefore the number individuals needed for the analysis is difficult to estimate.

To date, the minimum sequence difference between *csd* alleles needed to fully establish femaleness has been determined to amount to a 5 amino acid difference in the csd-PSD sequence. At the same time, it has been observed that the 3 amino acid difference in length in the HVR caused by a single indel mutation led in most cases to male development, but could also, although infrequently, determine femaleness [13]. In this study we pushed the boundary conditions for functional heterozygosity even further, to a virtually absolute minimum. One of the genotypes identified in our study, mB-p29, was characterised by a difference of 4 amino acids in the HVR region of the protein, but an additional single amino acid substitution was found in the RS-domain. However the second genotype, mB-p66, differed only by a single amino acid in the HVR length and no additional differences in the csd-PSD were found. The identification of the functional mB-p66 pair calls into question the validity of the current approach, i.e. establishing the criteria of functional heterozygosity based only on the number of differences in length and amino acid substitutions in the csd-PSD.

It can be argued that since we have genotyped the specimens at the pupa stage of their development, one can't exclude the possibility that the identified genotype mB-p66 still causes partial lethality due to incomplete penetrance of femaleness. This means however, given that no significant differences of the occurrence between mB-p66 and mA-p66 genotypes have been observed, the mA-p66 was also partially lethal. We find this scenario highly unlikely because the difference between alleles mA/p66 was much bigger (11 amino acid residues) than the previously established minimal criteria for functional heterozygosity (5 amino acid residues). It can still argue that the observed genotype frequencies could be explained by two independent causes: a) the mB-p66 genotype are not fully capable to determine femaleness due to too small sequence differences, b) and at the same time the mA-p66 genotype are partially lethal for reason independent from the *csd* genotype. Again, we think that this scenario is unlikely because the overall frequency of p66 allele occurrence in the analysed colony (3–2019) is higher than most of the other paternal alleles in this colony (5-th out of 18 in the order of frequency of the paternal alleles, Fig 3C) suggesting that both genotypes do not cause lethality.

Based on the evidence that, on the one hand, the 3 amino acid difference in HVR length is unable to fully determine femaleness (alleles G2-Y2 [13]), and, on the other hand, that the HVR length difference of 1 amino acid may be fully functional, it should be concluded that the criteria of functional heterozygosity are highly dependent on the position of the change in the

amino acid sequence. We also noticed that the difference between the mB-p66 allele pair is localised within the 3 amino acid difference characterizing the previously described G2-Y2 allele pair. Given that the mB-p66 vs. G2-Y2 sequence alignment shows significant differences in other csd-PSD positions, it should also be inferred that the difference causing functional heterozygosity depends not only on the position in the csd-PSD sequence, but also on the surrounding sequence context. In other words, the same or a similar mutation located at the same site may have a different effect depending on the allele in which it occurs. An additional indication for the dependence of functional heterozygosity on the context of the sequence and type of difference, but not necessarily the number of accumulated changes, is the identification of underrepresented *csd* genotypes, which in terms of the difference in HVR length and number of differences outside the HVR did not differ from other genotypes identified in this study. It should be noted, however, that the observed differences in the frequency of paternal and maternal allele pairing, are much smaller than those observed in a similar analysis by Beye et al. [13], where the effect of incomplete penetrance of femaleness led to the development of 1 female out of 45 embryos. Therefore, it remains to be established whether the underrepresentation of these genotypes is really caused by the *csd* configuration or other unfavourable genotypes linked with a given *csd* allele.

The discovery of *csd* genotype with minimal difference in the sequence of *csd*-PSD but fully functional in determining femaleness and showing that changes in the *csd* sequence have a different effect on generating functional heterozygosity depending on the context of the adjacent sequence is important for further research on the *csd* gene and its diversity. It becomes clear that until the exact structure and mechanism of action of the csd protein and its interactions with other proteins is fully characterised, predicting what difference in the sequence of the *csd* alleles (at least when minor differences are concerned) will be a functional one may be significantly flawed. This may be crucial for the analysis of the diversity of *csd* alleles and the subsequent estimation of the number of potentially functional *csd* genotypes present in a given population, as well as for predicting lethal genotypes in bee breeding programs [16, 17]. It may also be worth considering to refocus the research regarding csd on the N-terminal part of the protein (encoded by exons 2–5); although it has been shown to be subject to purifying selection and accumulates far less changes than the csd-PSD, at this point one cannot outright dismiss the possibility that very small differences in csd-PSD may be supported by more substantial differences in the N-terminal part of the protein to establish functional heterozygosity. Keeping that possibility in mind it may become necessary to sequence the full length of several hundred variants of *csd* identified so far on the basis of csd-PSD alone.

The discovery of the G2-Y2 *csd* pair differing only in HVR length and capable of partial female development determination led to the conclusion that the emergence of new *csd* specificities follows a path of increased fitness by increasing the penetrance of femaleness. It has been suggested that the fast-evolving length differences in the HVR are the initial step of the separation of *csd* specificities and are followed by subsequent mutations in other parts of the *csd*-PSD. This model assumes that no molecular intermediate forms that evolved through selectively neutral mutations are required [13]. While we find these conclusions still valid, at the same time, we show that the generation of a new, fully functional *csd* variant may also take place in a single mutation step involving a minimal change in the HVR length only, and therefore, we conclude that at least in some cases (depending on the position of the mutation and the sequence affected), no evolutionary intermediate forms of any kind may be needed to establish a new *csd* specificity. This means that the rate of emergence of new *csd* specificities in the population may be higher than expected. However, due to limited number of nearly identical (functional or partially functional) *csd* allele pairs identified so far, it is still unclear which evolutionary scenario is more likely to take place.

## Methods

### Simulation of the number of the restriction patterns of *csd* alleles present in spermathecas

All sequences used in the simulation were retrieved from GenBank (nucleotide) in May 2022 using the search parameters "*Apis mellifera*" in [organism] and "*csd*". Only records containing genomic DNA sequences were considered for further processing. Out of 667 retrieved records, 317 records contained sequences of appropriate length (flanked by csdF2 and csdR2 primers, see Fig 1). The retrieved sequences were trimmed to contain the csdF2-csdR2 fragment and then exported to MS Excel for further processing. Next, the duplicates were excluded and the resulting sequences (214 records) were processed to remove the intron between exons 7 and 8. The resulting sequences were translated *in vitro* and records with corrupted open reading frames were excluded from the dataset. As a result 151 genomic sequences flanked by csdF2 and csdR2 primers and containing the intron was used in the simulation. In each sequence the position of *VspI* restriction site (ATTAAT) has been localized and the length of the 5' and 3' fragments were established. Seventy nine unique *VspI* restriction patterns had been identified assigned to the records and their ID numbers. The records with their allele ID, corresponding restriction pattern ID, length of the 5' and 3' restriction fragments are available in the S1 File.

From this set, we created 10 subsets containing 80 randomly-picked distinct records (Kutools for MS Excel, "random data" tool)–the number of records in the subsets was chosen based on the reported *csd* diversity present in the local populations. Out of each of these subsets, "virtual spermathecas" containing 10, 12, 14, 16, 18 and 20 records were picked using RANDBETWEEN function. 10 virtual spermathecas were created for each subset and spermatheca size resulting in 100 simulations for each spermatheca size. The content of the virtual spermathecas were analysed to establish the number of distinct *csd* alleles (represented by the allele ID) and the number of unique corresponding restriction patterns (represented by the restriction pattern ID).

### Sample collection and DNA isolation

The worker brood was collected from five colonies headed by naturally inseminated 1-year-old queens in 2019 and 2020 (April) by cutting out a section of the capped brood comb. We have assumed that the queens mated with 15 drones and that the minimal size of the patriline group needed to identify potential biases in pairing with the maternal alleles within a given patriline is 16, thus we have estimated the number of individuals needed for the analysis as 240. The individual specimens were transferred into 48-well plates and stored in 96% ethanol at -20°C prior to further processing. The DNA was isolated using in-house-made silica-coated magnetic nanoparticles [18]. Briefly, ~20 mg of tissue was incubated in 60 ul of TNES buffer (100 mM Tris-HCl pH 8.0, 5 mM EDTA, 0.3% SDS, 200 mM NaCl) with 4 ul of proteinase K (10 mg/ml) for 2 hours at 56°C with shaking. Then 120 ul of GITC lysis buffer (6M guanidine isothiocyanate, 50 mM Tris-HCl pH 7.6–8.0, 20 mM EDTA, 2% sarkosyl) was added followed by precipitation with 240 ul of isopropanol. The DNA was then bound to magnetic particles which were settled on a magnetic rack. The DNA-bead complexes were washed once with 400 ul of isopropanol, twice with 300 ul of 80% ethanol, dried at 56°C, resuspended in 50 ul of nuclease-free water and incubated with shaking for 5 minutes at 56°C to enable DNA elution [18]. After settling the beads on the magnetic rack, 1 ul of the DNA solution was used for PCR.

### Terminal restriction length polymorphism

The Terminal Restriction Length Polymorphism analysis of *csd* gene was performed as following: the fragment encoding the potential-specifying domain of *csd* gene was amplified in a

two-step nested PCR reaction (30 cycles, 51˚C annealing) using the following set of primers [19]: csdF1: 5′AGACrATATGAAAAATTACACAATGA, csdR1: 5′TCATwTTTTCATTATTCA and csdF2: 5′-HEX-TATCGAGAAAsATCGAAAGAACGAT, csdR2: 5′-6FAM-ATTGAAAT CCAAGGTCCCATTGGT using PCR Mix Plus kit containing PCR antiinhibitors (A&A Biotechnology). 1–2 ul of the second amplification product was digested in 10 ul of reaction mixture containing 2.5 U of *VspI* restriction enzyme (Thermo Scientific) and incubated for 1 hour at 37˚C. 1 ul of the digestion product was denatured in the presence of Hi-Di formamide and 0.25 ul of 350 bp Rox standard (Life Technologies) and subjected to capillary electrophoresis on ABI Prism 310 apparatus. The size of the restriction fragments was assessed using Gene-Marker software.

## Cloning and sequencing

To clone the *csd* alleles from female specimens the sequence- and ligation-independent cloning (SLIC) method was used. The product of the first PCR reaction (see above) was reamplified using primers containing overhangs complementary to pUC-18 vector: SLIC-csdF2: 5′- AG GTCGACTCTAGAGGATCCTATCGAGAAAsATCGAAAGAACGAT and SLIC-csdR2: 5′- ATG ACCATGATTACGAATTCATTGAAATCCAAGGTCCCATTGGT. 60 ng of the gel-purified amplification product was then mixed with 150 ng of pUC-18 plasmid linearized with *EcoRI* and *BamHI* (Thermo Scientific) and the single-stranded overhangs were obtained in a 10 ul reaction containing T4 DNA polymerase (Thermo Scientific), 1x T4 DNA polymerase reaction buffer, 1 ul BSA (1 mg/ml) for 3 minutes at room temperature. The reaction was terminated by adding 1 ul of 10 mM dCTP and 20 ng of recA protein. The reaction mixture was then used for transformation of chemicompetent *E.coli* followed by ampicillin selection and blue/white screening. Single white colonies were picked and used directly in a PCR reaction (30 cycles, 52˚C annealing) using M13forward: 5′- CCCAGTCACGACGTTGTAAAACG and M13reverse: 5′- AGCGGATAACAATTTCACACAGG primers. The amplification product was digested with alkaline phosphatase (0.25 U, Thermo Scientific) and exonuclease I (0.5 U, Thermo Scientific) for 30 minutes at 37˚C. The enzymes were inactivated by denaturation (5 minutes, 95˚C). 1–3 ul of the DNA was used in cycle sequencing reaction (10 ul) containing: 1.9 ul 5x sequencing buffer, 0.5 ul BigDye Terminator Cycle Sequencing mix and 0.65 ul 5 uM of either csdF2 or csdR2 primer. The sequencing products were precipitated with 75% isopropanol in the presence of 0.25 ul of glycogen (10 mg/ml), resuspended in 15 ul of Hi-Di formamide, denatured and subjected to capillary electrophoresis (ABI Prism 310). Data were analyzed using ABI Sequence Analysis software (v3.3).

In individual cases, in order to clone fragments extending the *csd*-PSD the same cloning and sequencing techniques were used as described above except for different sets of primers for both nested-PCR reactions and sequencing (csdF0<->csdR1 [first PCR] and SLIC-csdF2<->SLIC-csdR1 [second PCR], or csdF0<->csdR0 [first PCR] and SLIC-csdF0<->SLIC-csdR2 [second PCR]) [20]. csdF0: 5′- GGGAGAGAAGTTGCAGTAGAG, csdR0: 5′- TTGATGCGTAGGTCCAAATCC, SLIC-csdR1: 5′- ATGACCATGATTACGAATTCGTCATCT CATwTTTTCATTATTCAAT, SLIC-csdF0: 5′- AGGTCGACTCTAGAGGATCCGGGAGAGAAG TTGCAGTAGAGATAGAAATAGAG.

The *csd* gene fragments derived from drones were amplified and sequenced without prior cloning.

## Data analysis

The worker bees within a given colony were assigned to paternal allele groups based on the nucleotide length of the restriction fragments of the paternal alleles. Within the paternal

groups, they were assigned to one of two maternal alleles present in the given colony. Testing for statistically significant differences between the expected frequencies of the presence of maternal alleles (0.5/0.5) and the observed frequencies within a given patriline is a problem of the simultaneous testing of more than one hypothesis i.e. testing of several patrilines within a colony. Therefore, Pearson chi-squared tests with Bonferroni correction for multiple (i.e. the number of tests performed for a given colony) testing (R statistical package) was used to adjust p-values to avoid the problem of higher probability of false rejecting the general null hypothesis, i.e. there is no significant differences in all considered cases.

In order to characterize sequence differences occurring in the allele pairs the obtained nucleotide sequences were *in silico* translated and aligned using Jalview pairwise alignment. The differences in the protein sequences were characterized by the length differences of the HVR and a total number of amino acid differences outside of the HVR.

### RNA isolation and RT-PCR

Total RNA was isolated from ~40 mg of tissue using Quick-RNA miniprep kit (Zymo Research) according to the manufacturer's recommendation. The RNA was then reverse-transcribed using Superscript IV reverse transcriptase (ThermoFisher Scientific) and random decamers and amplified in a PCR reaction (30 cycles, 52˚C annealing). In order to amplify *csd* transcripts, csdF1/csdR1 and csdF2/csdR2 were used as described above. In order to detect female- or male-specific splice forms of *dsx* gene the following primer sets were used: dsxM-A: 5′–TGGTCACCCATTTGCCACAGAC, dsxM-B: 5′–TCGTATGTCGGAGGTCCCGTTG (male) and dsxF-A: 5′–CTATTGGAGCACAGTAGCAAACTTG, dsxF-C: 5′–GAAACAATTTTGTT–CAAAATAGAATTCC (female). The amplification of *ef1-α* gene was used as a control of cDNA quality: ef1F: 5′–CGTTCGTACCGATCTCCGGATG, ef1R: 5′–GCTGCTGGAGCGAATGTTAC. The amplification products were resolved in 2% agarose gel.

### Supporting information

**S1 File. Sequences and record IDs of csd alleles retrieved from the public databases used in virtual validation of T-RFLP method.**
(XLSX)

**S2 File. Sequences and accession numbers of the csd alleles identified throughout this work.**
(XLSX)

**S3 File. Summary of the paternal csd allele identification in the analysed colonies and frequency of pairing with maternal alleles.**
(XLSX)

**S1 Raw images. Raw images used in Fig 6.**
(PDF)

### Acknowledgments

We would like to thank Ebenezer Christian, Oskar Fibich and Magdalena Lechowska for technical assistance and Pawel Mackiewicz for critical reading of the manuscript. We would like to dedicate this work in memory of our mentor, Stanislaw Cebrat.

## Author Contributions

**Conceptualization:** Malgorzata Cebrat.

**Data curation:** Robert Mroczek, Malgorzata Cebrat.

**Formal analysis:** Robert Mroczek, Pawel Blazej, Malgorzata Cebrat.

**Funding acquisition:** Malgorzata Cebrat.

**Investigation:** Robert Mroczek, Agnieszka Laszkiewicz, Kinga Adamczyk-Weglarzy, Joanna Niedbalska-Tarnowska, Malgorzata Cebrat.

**Methodology:** Robert Mroczek, Agnieszka Laszkiewicz, Malgorzata Cebrat.

**Project administration:** Malgorzata Cebrat.

**Resources:** Malgorzata Cebrat.

**Software:** Robert Mroczek.

**Supervision:** Malgorzata Cebrat.

**Validation:** Malgorzata Cebrat.

**Visualization:** Robert Mroczek, Joanna Niedbalska-Tarnowska.

**Writing – original draft:** Robert Mroczek, Malgorzata Cebrat.

**Writing – review & editing:** Malgorzata Cebrat.

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
