## [Decision Letter · Decision Letter 0]

22 Apr 2022

PONE-D-22-09346Revisiting the criteria of functional heterozygosity of the Apis mellifera complementary sex determining genePLOS ONE

Dear Dr. Cebrat,

Thank you for submitting your manuscript to PLOS ONE. After careful consideration, we feel that it has merit but does not fully meet PLOS ONE’s publication criteria as it currently stands. Therefore, we invite you to submit a revised version of the manuscript that addresses the points raised during the review process. As you can see from the accompanying critiques, one reviewer liked your manuscript as it is while the second one raises some points which you should address in a revised version. Since additional experiments are not required I estimate the extent of the requested changes anywhere between 'minor' and 'major'. 

We look forward to receiving your revised manuscript.

Kind regards,

Christoph Englert

Academic Editor

PLOS ONE

Journal Requirements:

Reviewers' comments:

Reviewer's Responses to Questions

**Comments to the Author**

1. Is the manuscript technically sound, and do the data support the conclusions?

Reviewer #1: Yes

Reviewer #2: Partly

2. Has the statistical analysis been performed appropriately and rigorously? 

Reviewer #1: Yes

Reviewer #2: N/A

3. Have the authors made all data underlying the findings in their manuscript fully available?

Reviewer #1: Yes

Reviewer #2: Yes

4. Is the manuscript presented in an intelligible fashion and written in standard English?

Reviewer #1: Yes

Reviewer #2: Yes

5. Review Comments to the Author

Reviewer #1: This well written manuscripts utilizes a new method to investigate the minimum sequence difference to result in functional CSD alleles that result in female honey bees. The finding that a single amino acid difference can result in femaleness is a significant breakthrough.

Line 80-81: There is no definition for dPSD, just dPSD

Reviewer #2: The manuscript report an interesting study on the A. mellifera csd gene - mainly focused on the identification of potential functional pairs.

The study is well written - however it needs several adjustemnts to better refine the content and clarify the approach

Introduction:

Line 47 - references should be reported. As far as I know, two publications list the number of csd alleles

Bilodeau, L., & Elsik, C. (2021). A scientific note defining allelic nomenclature standards for the highly diverse complementary sex-determiner (csd) locus in honey bees. Apidologie, 52(4), 749-754.

Bovo, S., Ribani, A., Utzeri, V. J., Taurisano, V., Schiavo, G., Bolner, M., & Fontanesi, L. (2021). Application of Next Generation Semiconductor-Based Sequencing for the Identification of Apis mellifera Complementary Sex Determiner (csd) Alleles from Honey DNA. Insects, 12(10), 868.

These works should be cited

The aim of the study is not completely clear - a methodology to enabling the analysis of a large number of the csd genotypes of worker bees in order to identify those that are underrepresented or absent in bee colonies, under the assumption that these genotypes induce, at least to some extent, lethality - was used but the purpose is not well defined as the results were not as the authors would have assumed

It seems that the study is divided in two part: one that would like to establish if there would be any underepresented genotypes and the other that analysed the minimum sequence differences between the pairs to define lethality or not.

The methodology should be however better defined

Results

Rationale and workflow of the genotyping method

A figure reporting the schematic representation of the methodology applied would help to follow the different steps and concepts

line 118: a figure reporting an example of the pattern observed with the T-RFLP (terminal restriction length polymorphism)-based method is needed to clarify the methodology and the way in which this approach was able to identify paternal alleles

Results were not really conclusive - this might be due to the fact that only five colonies were sampled - in addition it is not known when pupae (in which part of the year were sampled) - according to the ovideposition of the queen there would be potentially some biases in the distribution of the paternal alleles over the ovideposition period/years of the queen - this bias has not been taken into account to discuss the first part of the results - the biased genotypes

Some concluding remarks are missed - what is the final message derived by t he obtained results ? what are the limits of the approach? this should be reported in the discussion and conclusion as well as in the abstract

Methods

All elements related to the simulation study is missed - it is described partially in results - however, it is not clear how csd sequence data were retrieved, when, and how have been filtered - Actually a much larger number of alleles than those used in the simulation.

All different steps of the simulation part should be described here. It is not very clear how virtual spermateca have been constructed and the potential biases due to unequal sperm representation have been considered in the simulation

How multiple testing was considered ?

It is not clear if the authors defined the minimum number of sequences to be obtained from workers to infer with a sound statistical approach if there would be any biases in the paternal allele distribution - this part is quite naive

6. PLOS authors have the option to publish the peer review history of their article (what does this mean?). If published, this will include your full peer review and any attached files.

Reviewer #1: No

Reviewer #2: No

---

## [Author Response · Author response to Decision Letter 0]

19 May 2022

Attached please find our response to the Reviewers’ comments regarding our paper: “Revisiting the criteria of functional heterozygosity of the Apis mellifera complementary sex determining gene” which we would like to publish in PLOS ONE. I hope you will find that we have adequately responded to the comments, made appropriate corrections and that our paper can be now accepted for publication. 

Below please find our point-by-point response. 

Kind regards – Malgorzata Cebrat

Reviewer 1

1. Line 80-81: There is no definition for dPSD, just dPSD

Answer: the definition of dPSD has been included in line 79.

Reviewer 2

1. Line 47 - references should be reported. As far as I know, two publications list the number of csd alleles

Bilodeau, L., & Elsik, C. (2021). A scientific note defining allelic nomenclature standards for the highly diverse complementary sex-determiner (csd) locus in honey bees. Apidologie, 52(4), 749-754.

Bovo, S., Ribani, A., Utzeri, V. J., Taurisano, V., Schiavo, G., Bolner, M., & Fontanesi, L. (2021). Application of Next Generation Semiconductor-Based Sequencing for the Identification of Apis mellifera Complementary Sex Determiner (csd) Alleles from Honey DNA. Insects, 12(10), 868.

These works should be cited

Answer: The above-mentioned papers were cited as references 6 and 7, respectively (lines 45 and 108)

2. The aim of the study is not completely clear - a methodology to enabling the analysis of a large number of the csd genotypes of worker bees in order to identify those that are underrepresented or absent in bee colonies, under the assumption that these genotypes induce, at least to some extent, lethality - was used but the purpose is not well defined as the results were not as the authors would have assumed

It seems that the study is divided in two part: one that would like to establish if there would be any underepresented genotypes and the other that analysed the minimum sequence differences between the pairs to define lethality or not.

Answer: Indeed, we agree that the aim was not completely clear and the results were presented in a confusing way. The aim of this study was to acquire information about a large number of csd genotypes of diploid individuals and to identify, whenever possible, genotypes that are underrepresented or absent in bee colonies in order to verify/refine the criteria of functional heterozygosity of csd alleles. We have now clarified this in the Introduction (lines 89-94) and Results section (lines 115-118). Also, the Results section had been rearranged to remove the misleading impression of the aforementioned division and to emphasize the importance of identifying csd genotypes in analyzed colonies (subsection Identification and characterisation of csd genotypes).

3. A figure reporting the schematic representation of the methodology applied would help to follow the different steps and concepts

Answer: The figure showing the schematic representation of the methodology is included in the revised version of the manuscript as Figure 2.

4. line 118: a figure reporting an example of the pattern observed with the T-RFLP (terminal restriction length polymorphism)-based method is needed to clarify the methodology and the way in which this approach was able to identify paternal alleles

Answer: The requested figure is included in the revised version of the manuscript as Figure 3A.

5. Results were not really conclusive - this might be due to the fact that only five colonies were sampled - in addition it is not known when pupae (in which part of the year were sampled) - according to the ovideposition of the queen there would be potentially some biases in the distribution of the paternal alleles over the ovideposition period/years of the queen - this bias has not been taken into account to discuss the first part of the results - the biased genotypes

Answer: The pupae were collected in early spring (April) and they were offspring of 1-year-old naturally inseminated queens. This information is provided in the Methods section (lines 413-414). We agree that there are biases in the distribution of the paternal alleles over the ovideposition period/years of the queen (as it has been hypothesized – e.g. due to improper mixing of spermatozoa in the spermatheca, Brodschneider et al, 2012), however, we would like to draw the Reviewer's attention to the fact that we have not attempted to compare the frequency of occurrence of paternal alleles in the analysed colony but the potential biases in the frequency of pairing of a given paternal allele with the two maternal alleles present in the colony. Figuratively speaking, we did not pay attention to differences in the number of worker bees identified in the patrilines (differences in the heights of the bars in Fig 3C) but we have analyzed each patriline separately in order to establish whether there was a difference in pairing with the maternal alleles (comparing blue vs orange parts of each bar). This fact has been emphasized in lines 121-124 of the revised version of the manuscript.

6. Some concluding remarks are missed - what is the final message derived by t he obtained results ? what are the limits of the approach? this should be reported in the discussion and conclusion as well as in the abstract

Answer: The final message of the manuscript is the identification of a fully functional csd pair differing by one amino acid residue and the resulting conclusion that criteria for functional heterozygosity of the csd gene are highly dependent on the sequence context and position of the change and the generation of a new csd specificity may occur during a single mutation step with no need for evolutionary intermediates accumulating further mutations. This message has been now included in the Abstract (lines 26-30) and the final part of the Discussion (lines 379-385). The limitations of the method implemented in our research – low efficiency in identifying potential biases in the frequencies with which a given paternal allele pairs with the maternal alleles (approx. 50% of the patrilines had to be excluded from the analysis due to the insufficient number of individuals needed to draw statistically meaningful results) has been now presented in the Results and Discussion sections (lines 207-211 and 301-313, respectively).

7. All elements related to the simulation study is missed - it is described partially in results - however, it is not clear how csd sequence data were retrieved, when, and how have been filtered - Actually a much larger number of alleles than those used in the simulation. All different steps of the simulation part should be described here. It is not very clear how virtual spermateca have been constructed and the potential biases due to unequal sperm representation have been considered in the simulation

Answer: Details of the simulation have been now included in subsection “Simulation of the number of the restriction patterns of csd alleles present in spermathecas” of the “Methods” section (lines 388-410). a) Yes, it is true that the databases contain much more records on Apis mellifera csd allele sequences – our most recent search (May 2022) retrieved 667 records. Regrettably, because most of the research is focusing on the hypervariable region alone, only 317 records contained the full sequence needed for our analysis (depicted in Figure 1 of the manuscript). After elimination of duplicates (317->214) and the records containing obvious errors in sequencing (out of frame indels in exon sequences) we have now been able to identify 151 distinct csd alleles on the level of nucleotide sequence. Those 151 alleles represented 79 distinct restriction patterns. We have now redone our simulation using the updated set of sequences. b) Because the aim of this simulation was to estimate the difference between the absolute number of the distinct alleles present and the number of distinct restriction patterns they represent, in our opinion it is not necessary to consider biases in sperm representation in the spermatheca. In other words – assuming that the queen mated with a given number of drones we wanted to check how many distinct alleles and restriction patterns are present in her spermatheca but not how they will be distributed in the worker bees.

8. How multiple testing was considered?

Answer: Testing for statistically significant differences between the expected frequencies of the presence of maternal alleles (0.5/0.5) and the observed frequencies within a given patriline is a problem of the simultaneous testing of more than one hypothesis i.e. testing of several patrilines within a colony . Therefore, Pearson chi-squared tests with Bonferroni correction for multiple (i.e. the number of tests performed for a given colony) testing (R statistical package) was used to adjust p-values to avoid the problem of higher probability of false rejecting the general null hypothesis, i.e. there is no significant differences in all considered cases. 

This information has now been included in the manuscript (lines 473-479)

9. It is not clear if the authors defined the minimum number of sequences to be obtained from workers to infer with a sound statistical approach if there would be any biases in the paternal allele distribution - this part is quite naïve

Answer: We have assumed that the queens mated with approx. 15 drones and that the minimal size of the patriline group needed to identify potential biases in pairing with the maternal alleles within a given patriline as 16, thus we have estimated the number of individuals needed for the analysis as 240. This information has been now included in the Methods section (lines 414-417). We have not been able to predict the extent of the bias in sperm distribution and we admit in the revised version of the manuscript that this assumed number was too small, since many patrilines groups had to be excluded from the analysis due to the insufficient number of individuals. In the revised version of the manuscript we have also emphasized the usefulness of our approach in identifying genotypes in the analysed colonies.

---

## [Decision Letter · Decision Letter 1]

30 Jun 2022

PONE-D-22-09346R1Revisiting the criteria of functional heterozygosity of the Apis mellifera complementary sex determining genePLOS ONE

Dear Dr. Cebrat,

Thank you for submitting your manuscript to PLOS ONE. After careful consideration, we feel that it has merit but does not fully meet PLOS ONE’s publication criteria as it currently stands. Therefore, we invite you to submit a revised version of the manuscript that addresses the points raised during the review process.

The reviewer still has a number of suggestions that should improve your manuscript. Please address those issues before we can consider your manuscript for publication. 

We look forward to receiving your revised manuscript.

Kind regards,

Christoph Englert

Academic Editor

PLOS ONE

Journal Requirements:

Reviewers' comments:

Reviewer's Responses to Questions

**Comments to the Author**

1. If the authors have adequately addressed your comments raised in a previous round of review and you feel that this manuscript is now acceptable for publication, you may indicate that here to bypass the “Comments to the Author” section, enter your conflict of interest statement in the “Confidential to Editor” section, and submit your "Accept" recommendation.

Reviewer #2: All comments have been addressed

2. Is the manuscript technically sound, and do the data support the conclusions?

Reviewer #2: Partly

3. Has the statistical analysis been performed appropriately and rigorously? 

Reviewer #2: Yes

4. Have the authors made all data underlying the findings in their manuscript fully available?

Reviewer #2: Yes

5. Is the manuscript presented in an intelligible fashion and written in standard English?

Reviewer #2: Yes

6. Review Comments to the Author

Reviewer #2: The manuscript has been improved and now is much more interesting.

Anyway, a few additional improvements are still needed.

1) The title: based on the results obtained - jus one sequence that differ by one amino acid is a preliminary suggestion and and a confirmation of what is stated

2) The abstract is largely written including introductions to the context -no M&M info and just a few lines of results are reported - it should be better balanced - please take also into account that too bold conclusions should be better balanced.

3) In Table 2: They are not clear the the numbers corresponding to the "frequency of maternal alleles"

4) Not clear the nomenclature reported at line 232

7. PLOS authors have the option to publish the peer review history of their article (what does this mean?). If published, this will include your full peer review and any attached files.

Reviewer #2: No

---

## [Author Response · Author response to Decision Letter 1]

6 Jul 2022

Attached please find our response to the Reviewer comments regarding our paper: “New insights into the criteria of functional heterozygosity of the Apis mellifera complementary sex determining gene – discovery of a functional allele pair differing by a single amino acid” (former title “Revisiting the criteria of functional heterozygosity of the Apis mellifera complementary sex determining gene”) which we would like to publish in PLOS ONE. I hope you will find that we have adequately responded to the comments, made appropriate corrections and that our paper can be now accepted for publication. 

Below please find our point-by-point response. 

Kind regards – Malgorzata Cebrat

1. “The title: based on the results obtained - jus one sequence that differ by one amino acid is a preliminary suggestion and and a confirmation of what is stated”

Answer: The title has been changed to precisely describe our discovery and their possible impact on the matter of functional heterozygosity of csd gene.

2) “The abstract is largely written including introductions to the context -no M&M info and just a few lines of results are reported - it should be better balanced - please take also into account that too bold conclusions should be better balanced.”

Answer: The abstract has been rewritten to accommodate Reviewer’s suggestions – the introduction part has been shortened to a minimum needed to understand the context of our work, the M&M information has been incorporated into the description of the results and the conclusions are, in our opinion, better balanced. Please also note that the final sentence of the Discussion part has been added to better balance our conclusions.

3) “In Table 2: They are not clear the the numbers corresponding to the "frequency of maternal alleles"

Answer: An additional line has been added to the Table 2 legend in order to clarify the issue. We have also added the information which maternal alleles (A to G) were present in a given colony and how many worker bees carried it.

4) “Not clear the nomenclature reported at line 232”

Answer: The description of the genotypes have been changed (lines 237-239)

---

## [Editor Report · Decision Letter 2]

11 Jul 2022

New insights into the criteria of functional heterozygosity of the Apis mellifera complementary sex determining gene – discovery of a functional allele pair differing by a single amino acid

PONE-D-22-09346R2

Dear Dr. Cebrat,

We’re pleased to inform you that your manuscript has been judged scientifically suitable for publication and will be formally accepted for publication once it meets all outstanding technical requirements.

Kind regards,

Christoph Englert

Academic Editor

PLOS ONE
---

## [Editor Report · Acceptance letter]

29 Jul 2022

PONE-D-22-09346R2 

New insights into the criteria of functional heterozygosity of the *Apis mellifera* complementary sex determining gene – discovery of a functional allele pair differing by a single amino acid 

Dear Dr. Cebrat:

I'm pleased to inform you that your manuscript has been deemed suitable for publication in PLOS ONE. Congratulations! Your manuscript is now with our production department. 

Kind regards, 

on behalf of

Dr. Christoph Englert 

Academic Editor

PLOS ONE